# Anthropometric Profile of Soccer Players as a Determinant of Position Specificity and Methodological Issues of Body Composition Estimation

**DOI:** 10.3390/ijerph16132386

**Published:** 2019-07-05

**Authors:** César Leão, Miguel Camões, Filipe Manuel Clemente, Pantelis Theo Nikolaidis, Ricardo Lima, Pedro Bezerra, Thomas Rosemann, Beat Knechtle

**Affiliations:** 1Polytechnic Institute of Viana do Castelo, School of Sport and Leisure, Melgaço 4960-320, Portugal; 2Research Center in Sports Sciences, Health Sciences and Human Development (CIDESD), Vila Real 5001-801, Portugal; 3Exercise Physiology Laboratory, Nikaia 18450, Greece; 4School of Health and Caring Sciences, University of West Attica, Egaleo 12243, Greece; 5Institute of Primary Care, University of Zurich, 8091 Zurich, Switzerland; 6Medbase St. Gallen Am Vadianplatz, 9001 St. Gallen, Switzerland

**Keywords:** anthropometry, soccer, position, skinfold equations

## Abstract

The aim of the present study was (a) to describe the anthropometric profile of a large group of soccer players based on different age groups and their playing positions on the field, and (b) to examine the variations of body composition among adult soccer players using diverse equations based on skinfold thickness. A total of 618 Greek soccer players who were grouped by age (i.e., 12–14, 14–16, 16–18, and 18–37 years) and playing position (i.e., goalkeeper, defender, midfielder, and forward) were evaluated for weight, height, and skinfolds. The Pařízková formula was used to estimate the percentage of body fat. Furthermore, for players who were 18 years or older the Reilly and Evans formulas was used to estimate the percentage of body fat. Independent of the age, in this large sample, goalkeepers presented higher values for weight, height and the percentage of body fat estimation as compared with other field positions. An anthropometric pattern was observed in each tactical position, namely, across a specific age of increasing maturation process (14–16 years). With the Pařízková formula, we found a mean (SD) range of variation in the percentage of body fat estimation between 4.87 ± 1.46 and 5.51 ± 1.46 as compared with the Evans formula. The same pattern of differences was found when the Reilly equation was considered. In conclusion, we observed a position specificity of anthropometric characteristics across different age categories. Additionally, the same data supported different validated equations which resulted in large differences in the final outcome estimations.

## 1. Introduction

For a football team, there are many important factors for success and it is difficult to detach anthropometric and physiological characteristics as crucial factors in sports performance [1]. Nevertheless, evaluation of body composition in soccer players helps to improve their performance and evaluate applied training plan results [1], which is an important component of the athletes’ individualized and periodized training process [2].

Although there is an association between age and body composition, there does not seem to be consensus as to whether this relationship is positive or negative. It is possible to find declines in fat mass and increases in fat-free mass with advancing age [3], however, it is also possible to find the opposite in similar populations [4].

There is also a relationship between some anthropometric characteristics, namely the fat mass, and the risk for lesions [2]. In addition to the relationship with the risk of injury, it is also possible to find a relationship between fat mass and some physiological performance characteristics, such as speed and power [5]. With respect to this we know that a higher percentage of body fat (%BF) is negatively associated with velocity over 20 meters, an important determinant variable in the performance of soccer players [6].

Throughout an entire season we find differences in %BF but not necessarily in weight [7,8]. In addition, differences in %BF between player positions has already been found, which indicate a positional specificity [7]. It is important to take this issue into account when comparing %BF averages in soccer players. Therefore, evaluation of body composition in elite soccer players may help to improve performance and to track the results of the applied training regimens [1,9].

However, the assessment of body composition incorporates some difficulties. Each technique presents advantages, however, also has limitations [10]. We know that there is a wide range of methods used, without standardization [11], which lead to quite different results [12], making it often impossible to make comparisons between samples from different studies. Despite the validity of using equations based on skinfolds (SKF) as a method of assessing body composition, one of the assumptions is that the choice of the formula used has been validated in a similar population [11]. For this reason, the Reilly equation seems to be an obvious choice when it comes to conducting research on body composition in soccer players [13]. In addition, in the real-life context often the time available is limited. The Evans equation is validated in athletes and requires the evaluation of fewer skinfolds, and therefore it could be an interesting alternative for the technician placed in a team context [14].

Anthropometric measures are globally used in training monitorization as an important determinant of performance. However, the literature is scarce regarding epidemiological data in this specific context and methodological discussion regarding estimation procedures is needed to clarify the application of the practice. Accordingly, we aimed to (1) describe the anthropometric profile of a large group of soccer players, across different age groups and based on their playing positions in the field, and (2) examine the variations in estimations of body composition taking into account different validated equations.

## 2. Materials and Methods

### 2.1. Participants

A cross-sectional study was conducted among Greek amateur, semi-professional and professional soccer players. A total of 618 male soccer players from Greece, in good health and not injured at the time of the testing, presenting a mean (SD) age of 18.18 (4.78) years were assessed during the competitive seasons of 2008–2009, 2009–2010, and 2010–2011. After the initial assessment, the sample was distributed by age group (i.e., 12–14 years old, 14–16 years old, 16–18 years old, and 18–37 years old) and by playing position (i.e., forward, midfielder, defender, and goalkeeper (GK)) (Table 1).

We based the distribution of the athletes on the format of the national and international competitions, which typically use these subgroups. All those responsible for the education of players under the age of 18 approved the participation of the athletes in the study and all players of legal age completed consent to participate (written informed consent in both cases). The present study followed the recommendations for the study of humans in accordance with the Declaration of Helsinki [15] and was approved by the local ethics committee (number EPL2008/1).

### 2.2. Anthropometric Procedures

We conducted all the tests in the laboratory, between 2008 and 2011, on weekdays between 8:00 a.m. and 2:00 p.m. Body weight (HD-351, Tanita, Arlington Heights, IL, USA) and height (SECA, Leicester, UK) were assessed to the nearest 0.1 kg and 0.1 cm, respectively, according to the guidelines of the manufacturer. Three measurements of each variable were completed, with the mean value being recorded.

We measured 10 skinfolds (cheek, wattle, chest I, triceps, subscapular, abdominal, chest II, suprailiac, thigh and calf) with a skinfold caliper (Harpenden, West Sussex, UK), and %BF was calculated using the formula proposed by Pařízková [16]. Three measurements of each fold were performed in a rotation, using the mean value in mm for the sum of the 10 skinfolds. All measurements were realized by a qualified and experienced tester. In addition, the %BF was calculated using the formula proposed by Reilly [1] and the formula proposed by Evans [14] for athletes over 18 years old. All equations are shown below:

%BF by equation of Paryzkova: −41.32 + 12.59 × log_e_x, 

where, x was the sum of 10 skinfolds;

%BF by equation of Reilly: 5.174 + (0.124 × Thigh) + (0.147 × Abdominal) + (0.196 × Triceps) + (0.130 × Calf);

%BF by equation of Evans: 8.997 + 0.24658 × (Triceps + Abdominal + Thigh) − 6.343 × (gender) − 1.998 × (race), 

where, for gender 1 is male and 0 is female, and for race 0 is Caucasian and 1 is black.

### 2.3. Statistical Analysis

All data on the anthropometric characteristics were stratified by playing positions and age group. All results were expressed as mean values and standard deviations (mean ± SD), and the statistical analysis tests were computed at 0.05 level of significance (*p* = 0.05). The Shapiro–Wilk test was applied to ascertain the normal distribution of data. One-way ANOVA was used to analyze the anthropometric mean differences between groups, stratified by chronological age. The Bonferroni test was executed to pairwise analysis. Statistical analyses were performed using SPSS v23.0 statistical software (SPSS Inc., Chicago, IL, USA). The effect size (ES) d calculation for pairwise comparisons was executed using Cohen d. The following thresholds were used to classify the magnitude of standardized changes: 0.0–0.2 trivial, 0.2–0.6 small, 0.6–1.2 moderate, 1.2–2.0 large, and >2.0 very large. The confidence interval of d was set at 95%.

## 3. Results

Descriptive values by playing positions and chronological age group are summarized in Table 2. Regarding the playing position, there were statistically significant differences for %BF and the sum of skinfolds (*d* = 1.143, −0.443, 1.870 moderate effect; and *d* = 1.204, −0.500, 1.936 large effect), respectively, between the GKs and the midfielders for the age group 12–14 years, however, not for height and weight. These results contrast with the ones found for the age group 16–18 years, where there were differences for height and weight (*d* = 1.117, 0.382, 1.874 moderate effect; and *d* = 1.696, 0.931, 2.502 large effect), respectively, however, not for the %BF and the sum of skinfolds. In this group, there were differences between the GKs and midfielders regarding height, and between the GKs and all the other positions regarding weight. There were statistically significant differences in height, weight, %BF and sum of SKF for the age group 14–16 years (*d* = 0.733, 0.145, 1.331 moderate effect; *d* = 0.853, 0.262, 1.457 moderate effect; *d* = 0.545, −0.039, 1.136 small effect; and d = 0.630, 0.045, 1.224 moderate effect), respectively, and for the age group 18–37 years (*d* = 1.054, 0.618, 1.500 moderate effect; *d* = 0.962, 0.529, 1.404 moderate effect; *d* = 0.480, 0.058, 0.906 small effect; and *d* = 0.451, 0.029, 0.877 small effect), respectively. We detected differences between GKs and forwards in weight and height, respectively, for the age group 14–16 years (*d* = 0.843, 0.117, 1.600 moderate effect; and *d* = 0.972, 0.238, 1.743 moderate effect). For this group we also noticed differences among midfielders and defenders in %BF and sum of SKF (*d* = −0.588, −0.950, −0.23, small effect; and *d* = −0.548, −0.909, −0.192 small effect), respectively. For the age group 18–37 years, there were differences between GKs and defenders and midfielders in weight (*d* = 0.748, 0.322, 1.181 moderate effect; and *d* = 0.962, 0.529, 1.404 moderate effect), respectively, and between GKs and the other positions regarding height (*d* = 0.644, 0.220, 1.073 moderate effect; *d* = 1.054, 0.618, 1.500 moderate effect; and *d* = 0.778, 0.267, 1.303 moderate effect). Regarding %BF and sum of SKF we noticed differences between GKs and defenders (*d* = 0.601, 0.178, 1.030 moderate effect; and *d* = 0.632, 0.209, 1.062 moderate effect), respectively.

In summary, we observed that height and weight increase over all age groups, while %BF and sum of skinfolds decrease with increasing age in a statistically significant way. We noticed a trend in the pattern (Figure 1) relative to height and weight across all ages that showed goalkeepers as always the tallest, the heaviest, and the players with the highest %BF mass, and consequently, with the highest sum of skinfolds.

With the skinfolds evaluated, three formulas were used to estimate %BF in the group of players older than 18 years. Table 3 shows the calculated values for the different playing positions, with the effect size found for each position.

Taking into account the results obtained using different formulas, the pattern of %BF among positions remains constant, with GKs having the higher values and the defenders the lower values, independent of the formula used. Nevertheless, it is possible to observe that the absolute values of %BF are significantly different within position based on different formulas used (*p* < 0.001). Table 4 shows the mean differences between formulas and we observed the impact on the percentage of body fat estimation among adult athletes. Massive differences were observed when the Pařízková formula was used, with a mean (SD) range of variation of %BF between 4.17 ± 1.91 and 5.18 ± 1.99 as compared with the Reilly formula, and between 4.87 ± 1.56 and 5.51 ± 1.46 as compared with Evans.

## 4. Discussion

This cross-sectional data among a large group of soccer players showed a position specificity of the anthropometric characteristics across different age groups stages. Furthermore, and regarding the methodological issue of body composition estimation, supported in different validated equations between adult athletes, significant differences were found with an objective implication in practice data interpretations.

Youth growth follows a normal pattern for age [17]. However, differences in height, weight and body fat mass in relation to the playing position have already been described [18], noticing that there are significant differences during the development process that have an impact on playing position performance. The maturation state of young players has been shown as a selection factor, which leads to greater weight and height of the selected players as compared with the unselected ones [19], giving prominence to the discussion of the relative age and the potential impact on the future of these athletes.

The literature review regarding soccer players indicated significant differences in anthropometric measures across playing positions [1,7,8,13,20,21,22,23,24,25,26,27,28], as well between age categories [17,18,19,27,28,29,30,31,32,33]. This positional specificity does not seem to be unique to an age group, and regarding this knowing the pattern for the different positions allows practices to be individually tailored, in order to develop specific attributes and optimize the players’ performance.

It is likely to find anthropometric differences amongst positions throughout the development process, which demonstrated that the goalkeepers tend to be the heaviest, the tallest and the players with more %BF, while the midfielders are the reverse [20,21]. In a study of professional players, significant anthropometric differences were found between positions, similar to those found at the youth level, with goalkeepers being heavier, tallest, and with the highest body fat mass, followed by the defenders, the forwards and finally the midfielders [1]. These outcomes seem to suggest that, regardless of age, there is a selection pattern according to the anthropometric characteristics based on the specificity of the playing position in the field.

In our sample, throughout the development process, we saw increases in height and weight and decreases in %BF, in agreement with what was expected [16,25]. Regarding the values found, in particular for %BF, we saw that the values are slightly higher than those already described for other populations [19,23,25,29,34].

Consistent with the results from other studies [1,35], we found that regarding anthropometric characteristics of the different playing positions after the GKs, and in descending order of weight, height, and %BF, we have the defenders, the forwards and lastly the midfield players. It is worth noting that this division into positional lines is not used in all studies, which may modify the results of this relationship between different playing positions. Among the defensive line positions, there are significant anthropometric differences between central and lateral defenders, which may change the results of this relationship between different playing positions [5].

Soccer is characterized by different physiological needs in the many field positions, which leads to different physical characteristics [36,37]. Regarding this, after the analysis of our results, we conclude that the goalkeeper position is the position where athletes have a greater weight, greater height, and higher %BF as compared with the other positions. We found that the mean for the height of our sample is lower than would be expected, considering the characteristics described by Ziv et al. [38] for the goalkeeper. This is explained by the fact that the athletes were studied from different ages, i.e., 12 years old up to above 18 years old. Nevertheless, this circumstance does not explain the difference in %BF, which was much higher than what is considered to be normal.

Therefore, a comparison between values should always be made with caution. One possibility may be to use the sum of the skinfolds, as proposed by ISAK [39], that show a good correlation with all methods of evaluation of %BF, and using it as an indicator of the athlete’s adiposity and its changes over time [40]. All methods of assessing body composition have their advantages and limitations [8,41], but normalizing the method used to evaluate soccer players in future studies may facilitate a comparison between different samples.

We note some reasons for the differences found between previous studies and our sample regarding the %BF, namely, the soccer level of the different samples in the different studies. It is assumed that the higher the level of soccer, the lower the %BF. In addition, different methods of estimating the %BF may lead to different results, and it is proven that the difference is not negligible [12]. The use of equations to calculate %BF has a good correlation with standard gold methods, such as DXA, but the different choices can increase the difference between methods [40].

For this reason, we calculated %BF values using different formulas for a subsample of adult players. Although it was not possible to make comparisons with a reference method, our data shows that the formula chosen to calculate the %BF has an impact on the final value found, which leads to different conclusions. Regardless of whether they are validated and have a good correlation with reference methods, the use of different equations to estimate the %BF does not permit values comparable to each other to be obtained [42,43].

Although we found more than 100 equations validated for estimating %BF, we noticed that when the population used for validation is different it may lead to differences between them. Furthermore, the fact that there are different variables used, such as the total number of different skinfolds instead of using skinfolds from different locations, also contributes to this disagreement [10].

### Limitations

One of the limitations of our study is the division of players into four horizontal positional lines, which was the most common division found in the literature. Nevertheless, for future research, we think a division into vertical lines could prove more helpful, leading to distinct results between the players in positions that are near the center and those who are closer to the lines [5]. Another limitation is the lack of data regarding the maturity level of the young soccer players, since this could provide better information about the differences between categories and if the differences between positions are due to the position specificity or whether the young players are chosen for the position because of their advanced maturity level relative to their peers.

## 5. Conclusions

Our study increases the knowledge about anthropometric characteristics of young and adult soccer players and about the evolution of these characteristics throughout the normal development process. In summary, age shows an increase in height and weight, while %BF shows a decrease, within the ranges that would be expected. In addition, an anthropometric pattern was observed in each tactical position, namely across a specific age of increasing maturation process (14–16 years). This seems to suggest that the player selection process already takes into account specialization in a position from an early age, and it seems that selection is manifested in later stages of development. Longitudinal data is needed to better clarify the prospective evolution of these characteristics among the athletes.

In addition, although under standardized procedures of assessment, different validated equations for estimating body composition result in huge differences in the final outcomes, with practical implications for monitoring the training process. Therefore, the sum of skinfolds or equations previously validated among similar target populations should be chosen.

## Figures and Tables

**Figure 1 ijerph-16-02386-f001:**
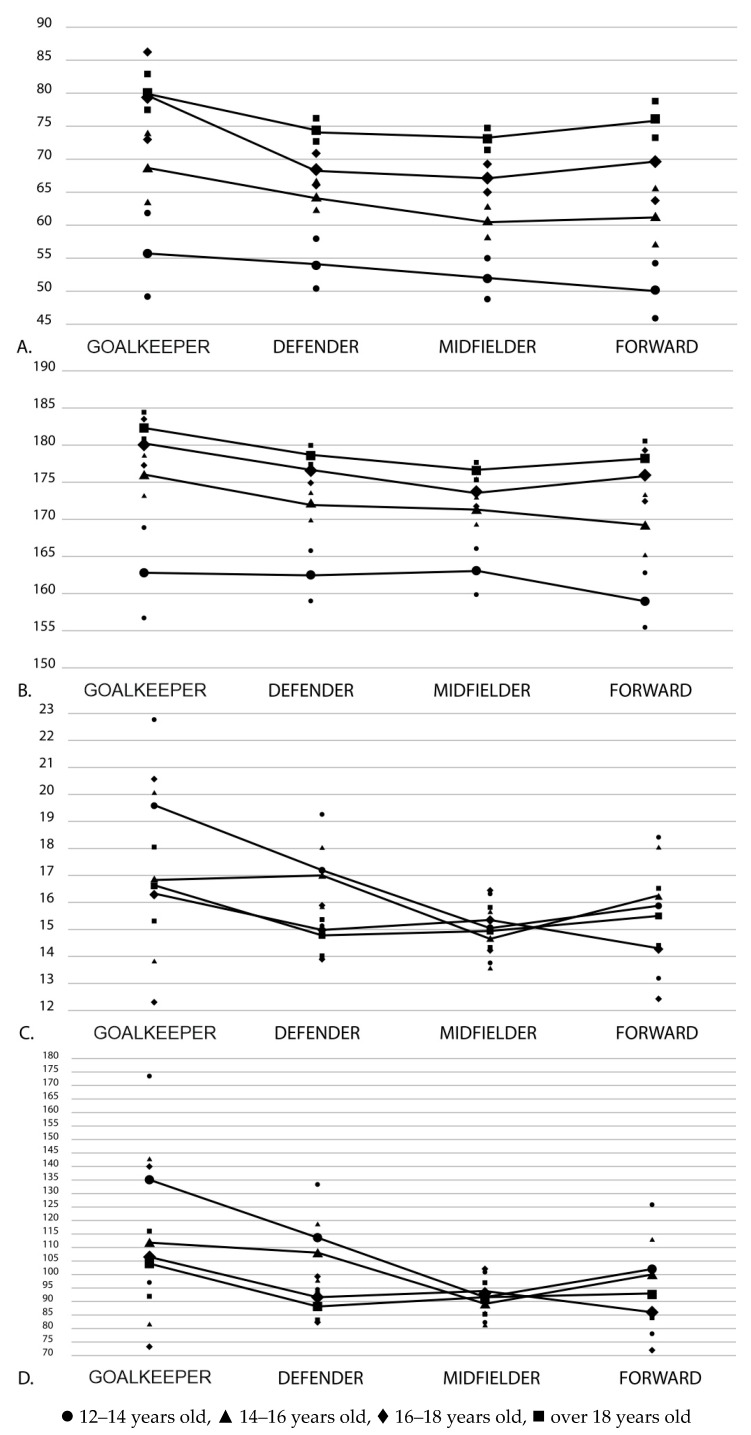
Means (CI 95 %) of the anthropometric variables (**A**) weight; (**B**) height; (**C**) % body fat; (**D**) sum of skinfolds) by age group and playing position.

**Table 1 ijerph-16-02386-t001:** Distribution by age group and playing position of the participants.

Age Group (Years old)	*n* (%)	Playing Position	*n* (%)
12–14	97 (15.7%)	Goalkeeper (GK)	63 (10.2%)
14–16	155 (25.1%)	Defender	237 (38.3%)
16–18	126 (20.4%)	Midfielder	232 (37.5%)
18–37	240 (38.8%)	Forward	86 (13.9%)

**Table 2 ijerph-16-02386-t002:** Descriptive values by chronological age group and playing position.

Age Group (years)	Position	Weight (kg)	*p*	Height (cm)	*p*	Body fat (%)	*p*	Ʃ SKF (mm)	*p*
12–14 years	Goalkeeper	55.71 ± 9.27	0.376	162.96 ± 8.98	0.455	19.61 ± 4.83	0.033	135.76 ± 57.72	0.23
	Defender	54.04 ± 10.14		162.50 ± 9.03		17.14 ± 5.42		114.19 ± 51.96	
	Midfielder	52.00 ± 9.06		163.10 ± 9.25		15.07 ± 3.61		91.94 ± 26.69	
	Forward	50.33 ± 8.95		159.23 ± 7.54		15.74 ± 5.42		102.21 ± 49.25	
14–16 years	Goalkeeper	68.78 ± 8.99	0.006	176.27 ± 5.44	0.052	16.91 ± 5.46	0.013	111.73 ± 52.44	0.018
	Defender	64.51 ± 8.12		172.10 ± 7.37		16.99 ± 4.16		108.78 ± 39.55	
	Midfielder	60.66 ± 9.51		171.29 ± 6.97		14.64 ± 3.77		89.33 ± 30.17	
	Forward	61.34 ± 8.27		169.37 ± 7.91		16.13 ± 3.60		99.58 ± 26.55	
16–18 years	Goalkeeper	79.76 ± 8.23	0.001	180.41 ± 4.20	0.007	16.42 ± 5.48	0.516	106.56 ± 43.28	0.400
	Defender	68.44 ± 8.61		176.90 ± 6.09		14.92 ± 3.83		91.42 ± 30.35	
	Midfielder	67.17 ± 7.13		173.69 ± 6.19		15.34 ± 3.63		94.01 ± 28.34	
	Forward	69.80 ± 11.67		176.02 ± 6.36		14.18 ± 3.63		85.72 ± 27.34	
Over 18 years	Goalkeeper	80.10 ± 7.11	0.000	182.69 ± 4.32	0.000	16.69 ± 3.59	0.044	104.36 ± 30.33	0.041
	Defender	74.50 ± 7.54		178.86 ± 6.33		14.69 ± 3.21		88.46 ± 23.04	
	Midfielder	73.39 ± 6.87		176.70 ± 6.01		15.01 ± 3.44		91.39 ± 27.69	
	Forward	76.10 ± 8.53		178.31 ± 6.45		15.35 ± 3.06		92.97 ± 24.04	

**Table 3 ijerph-16-02386-t003:** %BF estimated using different equations by playing position for players over 18 years old.

**Position**	**%BF Pařízková (%)**	**%BF Reilly (%)**	***p***	***Cohen d***
Forward	15.35 ± 3.06	10.66 ± 1.43	<0.001	1.9754
Midfielder	15.01 ± 3.44	10.73 ± 1.85	<0.001	1.5576
Defender	14.69 ± 3.21	10.52 ± 1.46	<0.001	1.6826
GK	16.69 ± 3.59	11.51 ± 1.80	<0.001	1.8343
**Position**	**%BF Pařízková (%)**	**%BF Evans (%)**	***p***	***Cohen d***
Forward	15.35 ± 3.06	10.12 ± 2.29	<0.001	1.9403
Midfielder	15.01 ± 3.44	10.09 ± 2.54	<0.001	1.8737
Defender	14.69 ± 3.21	9.82 ± 1.97	<0.001	2.1572
GK	16.69 ± 3.59	11.18 ± 2.43	<0.001	1.5124

**Table 4 ijerph-16-02386-t004:** Mean difference between equations used to calculated %BF by player position.

Position	Pařízková–Reilly	Pařízková–Evans	Reilly–Evans (%)
Forward	4.69 ± 1.84	5.28 ± 1.55	0.59 ± 0.71
Midfielder	4.29 ± 1.79	4.92 ± 1.35	0.64 ± 0.74
Defender	4.17 ± 1.91	4.87 ± 1.56	0.70 ± 0.58
GK	5.18 ± 1.99	5.51 ± 1.46	0.33 ± 0.71

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
