# Peer review of "Anthropometric Profile of Soccer Players as a Determinant of Position Specificity and Methodological Issues of Body Composition Estimation"

_ijerph, 2019, doi:10.3390/ijerph16132386_

Round 1
Reviewer 1 Report
General Comments
The article “Anthropometric profile of soccer players as determinant of position specificity: a cross-sectional study” presents anthropometric data of players from different age and positions, and also discuss different equations for estimating body fat mass of soccer players. It provides interesting insights into the area and a significant contribution. I highlight the fact that playing position differences are less prominent in the younger group, which indicates a strong impact of training process on professional players’ anthropometric characteristic (what could be more explored by the authors). However, I point out some issues regarding the style of writing and require some methodological explanations, which are presented below.
Specific comments
Abstract
Line 25: Goalkeepers, Defenders, Midfielders and Forwards (please, adjust).
Line 25: body fat percentage or percentage of body fat (the same in line 27)
Line 28: “We have …. Differences for height, height” (something is wrong here).
Introduction
Although it provides some interesting points about the main topics of the study, there is no rationale in regard to the research problem. In my opinion, the introduction should be completely rewritten considering: a) what is the research problem addressed by this manuscript (Is it the use of different equations? Is it the positional differences? Is it the playing position effect?) b) What is the relevance of addressing this issue? c) Based on the literature, what results are expected?
For example: there is nothing about age-related effects in the introduction, although this is a major issue addressed by the manuscript.
Materials and methods
Playing position classification must be better specified. Fullbacks were classified as defenders (since fullbacks and center-backs have a completely different anthropometric characteristic?) Wings were classified as forwards or midfielders? I recommend you to include a figure with the positions and classifications (See DiSalvo, 2007).
Since you have two main factors (playing position and age), I am not sure if one-way ANOVA is the most recommended test. I would consider using a two-way ANOVA (playing position x age), since you may have interactions between main effects that are not captured by the one-way analysis of variance. What do you think about it?
For comparisons between groups, it is mandatory to include the Effect Size.
Are you sure that the regression was used to measure associations? Does the regression provide only this?
Results
Figure 1: You used GK for goalkeeper, but you did not use initials for the other playing positions. I recommend you to standardize it.
Discussion
Lines 170-177: If this is a conclusion, it is not supposed to be here.
I would recommend you to better organize information throughout this section. There is no coherence between the paragraphs. It would be great if you are able to put the information in the same order in both introduction and discussion (for example differences between equations, the effect of playing position and effect of age). It would make the article much more comprehensible for the readers.
Author Response
Comments and Suggestions for Authors
General Comments
The article “Anthropometric profile of soccer players as determinant of position specificity: a cross-sectional study” presents anthropometric data of players from different age and positions, and also discuss different equations for estimating body fat mass of soccer players. It provides interesting insights into the area and a significant contribution. I highlight the fact that playing position differences are less prominent in the younger group, which indicates a strong impact of training process on professional players’ anthropometric characteristic (what could be more explored by the authors). However, I point out some issues regarding the style of writing and require some methodological explanations, which are presented below.
Specific comments
Abstract
Line 25: Goalkeepers, Defenders, Midfielders and Forwards (please, adjust).
Answer: We agree with the expert reviewer and we made the suggested adjustments
Line 25: body fat percentage or percentage of body fat (the same in line 27)
Answer: We agree with the expert reviewer and we made the suggested adjustments
Line 28: “We have …. Differences for height, height” (something is wrong here).
Answer: We agree with the expert reviewer and we made the necessary adjustment
Introduction
Although it provides some interesting points about the main topics of the study, there is no rationale in regard to the research problem. In my opinion, the introduction should be completely rewritten considering: a) what is the research problem addressed by this manuscript (Is it the use of different equations? Is it the positional differences? Is it the playing position effect?) b) What is the relevance of addressing this issue? c) Based on the literature, what results are expected?
Answer: We agree with the expert reviewer and the introduction section was redesigned highlighting clearly the specific objectives. The problem was addressed accordingly the literature; first the question of the anthropometric profile and his specificity in the field and second clarifying the methodological problem of estimation body composition.
For example: there is nothing about age-related effects in the introduction, although this is a major issue addressed by the manuscript.
Answer: We agree with the expert reviewer and specific data were added to the introduction section.
Materials and methods
Playing position classification must be better specified. Fullbacks were classified as defenders (since fullbacks and center-backs have a completely different anthropometric characteristic?) Wings were classified as forwards or midfielders? I recommend you to include a figure with the positions and classifications (See DiSalvo, 2007).
Answer: We agree with the expert reviewer and in reality this is one of the limitations we find in the literature. There is no uniformity of the field positions in the written articles, at the time of data collection we used the distribution used by several authors, such as Malina 2001, Matkovic 2003, Reilly 2009 and Sutton 2009, among others.
Since you have two main factors (playing position and age), I am not sure if one-way ANOVA is the most recommended test. I would consider using a two-way ANOVA (playing position x age), since you may have interactions between main effects that are not captured by the one-way analysis of variance. What do you think about it?
Answer: One of the assumptions of two-way ANOVA cannot be assumed in which we should have independence of observations, which means that there is no relationship between the observations in each group or between the groups themselves. Based on that, we can’t perform the two-way ANOVA.
For comparisons between groups, it is mandatory to include the Effect Size.
Answer: We agree with the expert reviewer and include the effect size in the results
Are you sure that the regression was used to measure associations? Does the regression provide only this?
Answer: We agree with the expert reviewer and we withdraw the following description from the main manuscript: “Multivariate regression analysis was conducted to test the associations between anthropometric characteristics, age and tactical positions.”
Results
Figure 1: You used GK for goalkeeper, but you did not use initials for the other playing positions. I recommend you to standardize it.
Answer: We agree with the expert reviewer and we made the suggested adjustments
Discussion
Lines 170-177: If this is a conclusion, it is not supposed to be here.
Answer: We agree with the expert reviewer and we made the necessary adjustment to the discussion and conclusion section.
I would recommend you to better organize information throughout this section. There is no coherence between the paragraphs. It would be great if you are able to put the information in the same order in both introduction and discussion (for example differences between equations, the effect of playing position and effect of age). It would make the article much more comprehensible for the readers.
Answer: We agree with the expert reviewer and the discussion section was revised to improve coherence around the first descriptive objective and about the methodological issue of body composition estimation.
Reviewer 2 Report
Abstract
Line 28 – height is listed twice.
Introduction
Lines 49-50 – some rewording of this sentence is needed.
Materials and methods.
Table 1 – it is not clear why younger goal keepers have been represented in this table. It is hard to compare anything as this seems to be a selection of playing positions within different age categories, what is the rationale for 12-14 year goalkeepers, 14-16 year defenders etc.?
Within the sample which was taken across 3 seasons, were there some of the same participants? Would this change the usefulness of the data? What about adjusting for adolescent body fat differences? This has not clearly been justified.
Line 101 – need clarification for a mixed gender sample, were the sample mixed? This would surely confound the results?
All research shows difference in height and weight for keepers in relation to other positions, why is it necessary to repeat this?
If the data is deemed necessary then it would be good to compare this with an ‘ideal’ or template for youth development and growth, i.e. what is normal.
It may be that the main focus of this work is the comparison of the different equations, if this is the case, then more should be made of this in the work and the introduction. It would then not be a case of comparing different playing positions but comparing the results from the equations and justifying the use of specific equations based on the results.
Line 200 - if the differences were already known then justification for using a more heterogeneous sample should be made.
Author Response
Comments and Suggestions for Authors
Abstract
Line 28 – height is listed twice.
Answer: We agree with the expert reviewer and we made the suggested adjustments
Introduction.
Lines 49-50 – some rewording of this sentence is needed.
Answer: We agree with the expert reviewer and we made the suggested adjustments
Materials and methods.
Table 1 – it is not clear why younger goal keepers have been represented in this table. It is hard to compare anything as this seems to be a selection of playing positions within different age categories, what is the rationale for 12-14 year goalkeepers, 14-16 year defenders etc.?
Answer: We agree with the expert reviewer and we would like to clarify that table 1 only aims to show the distribution of the sample in its entirety by the age groups as well as by the tactical positions in the game. Table 1 main objective is to describe only the dimensions of the sample regarding both major variables in analyze: age and playing position.
Within the sample which was taken across 3 seasons, were there some of the same participants? Would this change the usefulness of the data? What about adjusting for adolescent body fat differences? This has not clearly been justified.
Answer: We agree with the expert reviewer and we would like to clarify that this was a cross-sectional study, with the participants of which all different over the three seasons. In relation to adjusting for differences in body fat in adolescence, we include a paragraph in the introduction that attempts to elucidate the relevance of the study.
Line 101 – need clarification for a mixed gender sample, were the sample mixed? This would surely confound the results?
Answer: We agree with the expert reviewer but we only describe the Evans equation in the methods. This equation have gender as co-variable in the final estimation. However, in our sample and specifically regarding this objective our sample only have males. The gender description was added to the participants section.
All research shows difference in height and weight for keepers in relation to other positions, why is it necessary to repeat this?
Answer: We agree with the expert reviewer and we only want to emphasize that these differences remain throughout the entire development process
If the data is deemed necessary then it would be good to compare this with an ‘ideal’ or template for youth development and growth, i.e. what is normal.
Answer: We agree with the expert reviewer issue of highlighting in discussion section the cross comparisons; however published data are scarce across different age groups and the methodological issues of measuring and final estimation of anthropometric outcomes, as shown in this specific manuscript, conducts large limitations in cross comparisons.
It may be that the main focus of this work is the comparison of the different equations, if this is the case, then more should be made of this in the work and the introduction. It would then not be a case of comparing different playing positions but comparing the results from the equations and justifying the use of specific equations based on the results.
Answer: We understand the expert reviewer doubt. However we believe that both objectives were relevantly highlighted and the second one complements the first. Firstly the cross sectional large description data and then the methodological issue of measuring and conducting to final estimation of sports determinants, that conducts to huge limitations in comparisons between observational studies. Specifically we believe that the discussion between topics conducts to an important improvement in the original manuscript.
Line 200 - if the differences were already known then justification for using a more heterogeneous sample should be made.
Answer: We agree with the expert reviewer and the differences found may be justified by these factors, but it was only after analyzing the data that we were able to compare.
Round 2
Reviewer 2 Report
The changes made in explaining the methods, data and the significance of the work make this article eminently more accessible to the reader.
Thank you for the corrections.
This manuscript is a resubmission of an earlier submission. The following is a list of the peer review reports and author responses from that submission.